# The yielding transition in amorphous solids under oscillatory shear deformation

Premkumar Leishangthem[1], Anshul D.S. Parmar[1,2] & Srikanth Sastry[1]

Amorphous solids are ubiquitous among natural and man-made materials. Often used as structural materials for their attractive mechanical properties, their utility depends critically on their response to applied stresses. Processes underlying such mechanical response, and in particular the yielding behaviour of amorphous solids, are not satisfactorily understood. Although studied extensively, observed yielding behaviour can be gradual and depend significantly on conditions of study, making it difficult to convincingly validate existing theoretical descriptions of a sharp yielding transition. Here we employ oscillatory deformation as a reliable probe of the yielding transition. Through extensive computer simulations for a wide range of system sizes, we demonstrate that cyclically deformed model glasses exhibit a sharply defined yielding transition with characteristics that are independent of preparation history. In contrast to prevailing expectations, the statistics of avalanches reveals no signature of the impending transition, but exhibit dramatic, qualitative, changes in character across the transition.

[1] Jawaharlal Nehru Center for Advanced Scientific Research, Jakkur Campus, Bengaluru 560064, India. [2] TIFR Center for Interdisciplinary Sciences, 21 Brundavan Colony, Narsingi, Hyderabad 500075, India. Correspondence and requests for materials should be addressed to S.S. (email: sastry@jncasr.ac.in).

The mechanical response to applied stresses or deformation is a basic material characteristic of solids, both crystalline and amorphous. Whereas the response to small perturbations are described by elastic moduli, the plastic, irreversible, response to large deformation[1–16] is often more important to characterize, as it determines many material parameters such as strength and ductility, and is also of relevance to thermomechanical processing of metallic glasses[17]. Amorphous solids lack the translational symmetry of crystals, and thus no obvious analogs to dislocation defects in terms of which plasticity in crystals has been sought to be understood. Based on work over the last decades, it is appreciated that plasticity arises in amorphous solids through spatially localized reorganizations[1,2,18], termed shear transformation zones, and that such localized zones interact with each other through long ranged elastic strains they induce[19]. While many details of the nature of these localized regions of non-affine displacements remain to be worked out, they form the basis of analyses and models of *elasto*-plasticity and yielding[7,15,19–21]. In addition to extensive experimental and theoretical investigations, computer simulations of atomistic models of glasses have also been employed, to eludictate key features of plastic response[1,2,4] on atomic scales. While several studies have been conducted at finite shear rates (for example,[13,15]), many studies have focussed on behaviour in the athermal, quasi-static (AQS)[4,6,7,16,22] limit, wherein the model glasses studied remain in zero temperature, local energy minimum, configurations as they are sheared quasi-statically. The AQS protocol represents a limit in which the deformation behaviour of the solids does not depend crucially on thermally induced processes, and relaxation processes are expected to occur on time scales much faster than the shear rate. Thus, results from AQS may be expected to be useful in understanding the behaviour of glasses sufficiently below the glass transition, and for small shear rates. Both these conditions may be expected to be satisfied in the context of understanding the mechanical failure of glasses. Such deformation induces discontinuous drops in energy and stress with corresponding nonaffine displacements that are highly spatially correlated, and exhibit power law distributions in size. In analogy with similar avalanches that arise in diverse context of intermittent response in disordered systems, from earthquakes, crackling noise in magnetic systems, depinning of interfaces in a disorded medium and so on[23], a theoretical description of mechanical failure in amorphous solids[5], predicts the mean avalanche size to diverge as a critical stress is approached from below, leading to a power law distribution with a diverging mean size at and above the transition. Indeed, it has been observed that (for example,[6,15,22]) system spanning avalanches are present in the steady state beyond yield, whose sizes scale with system size. The character of avalanches upon approaching the yielding transition, however, has not received much attention, as also the differences between pre- and post-yield avalanches. Among the reasons is the sample to sample variability of behaviour below yield, in contrast with the universal behaviour seen in the post-yield regime.

Here we show that oscillatory deformation offers a robust approach to systematically probe behaviour above and below a sharply defined point of mechanical failure, which we associate with yielding. As our results pertain to oscillatory deformation in the limit of vanishing shear rate, we caution that comparisons with uniform shear at finite rates must be made with due care. Oscillatory deformation is a widely used experimental technique[12,24–29] as well as a common protocol in materials testing. However, it has not been employed widely in computational investigations, barring some recent work[14,30–32], to probe yielding in amorphous solids. In the present work, we perform an extensive computational study of plastic response in a model glass former, over a wide range of system sizes, and amplitudes of deformation that straddle the yielding strain.

## Results

**Simulations**. We study the Kob-Andersen 80:20 binary mixture Lennard-Jones glasses for a range of system sizes (see Methods for details). The glasses studied are prepared by performing a local energy minimization of equilibrated liquid configurations, at a reduced temperatures $T = 1$ and $T = 0.466$. The inherent structures so obtained represent poorly annealed ($T = 1$) and well annealed ($T = 0.466$) glasses. These glasses, referred to by the corresponding liquid temperature in what follows, are subjected to volume preserving shear deformation through the AQS protocol, wherein the strain $\gamma_{xz}$ is incremented in small steps, with each step being followed by energy minimization. The strain is incremented in the same direction in the case of uniform strain, whereas for oscillatory strain for a given maximum amplitude $\gamma_{max}$, a cycle of strain $0 \rightarrow \gamma_{max} \rightarrow 0 \rightarrow -\gamma_{max} \rightarrow 0$ is applied repeatedly over many cycles, until a steady state is reached. Results presented below, except Fig. 1d are from analysing steady state configurations. Further details concerning the simulations and analysis are presented in Methods and Supplementary Figs 1–11.

**Yielding transition**. Previous work[30] has shown that as the amplitude of strain $\gamma_{max}$ approaches a critical value $\gamma_y$ from either side, the number of cycles needed to reach the steady state becomes increasingly large, with an apparent divergence at $\gamma_y$ (Supplementary Fig. 1). We identify $\gamma_y$ ($\sim 0.07$) as the yield strain, as justified below. In Fig. 1a we show the averaged stress–strain curves for $N = 4,000$. For each $\gamma_{max}$, we obtain a maximum stress $\sigma_{max}$ reached at $\gamma = \gamma_{max}$, which are plotted in Fig. 1b for $T = 1$, 0.466, for $N = 4,000$, 32,000. Figure 1b also shows the stress–strain curves for the same cases obtained with uniform strain. Whereas stresses vary smoothly for uniform strain, with no sharp signature of the onset of yielding, and differ significantly for $T = 1$ and $T = 0.466$, they display a sharp, discontinuous, drop above $\gamma_{max} = 0.07$ (0.08 for $N = 4,000$) for oscillatory strain. Interestingly, below $\gamma_y$, the maximum stress increases as a result of oscillatory deformation, indicative of hardening, consistently with previous results[33]. Above $\gamma_y$, repeated oscillatory deformation leads to a stress drop relative to values just below $\gamma_y$, indicating yielding.

Figure 1c displays the potential energies obtained over a full cycle in the steady state (Supplementary Fig. 2). For $\gamma_{max} < \gamma_y$, the energies display a single minimum close to $\gamma = 0$, but above, bifurcate into two minima, indicating the emergence of plasticity. The stress–strain curves show a corresponding emergence of loops (Fig. 1a) with finite area. Strain values at the minima for energy, $\gamma_{Umin}$ and $\sigma_{xz} = 0$, $\gamma_{\sigma_0}$ (see Supplementary Fig. 3), are shown in Fig. 1d as a function of the number of cycles for different $\gamma_{max}$. We note that $\gamma_{max} = 0.08$ displays interesting non-monotonic behaviour, with an initial decrease in these strain values, similar to smaller $\gamma_{max}$, but an eventual increase to larger strains, similar to the case $\gamma_{max} = 0.12$, in the yielded regime. Figure 1e shows $\gamma_{Umin}$ and $\gamma_{\sigma_0}$ versus $\gamma_{max}$, which show an apparently continuous departure from nearly zero, signalling a transition at $\gamma_{max} \sim 0.07$. Figure 1f shows that the minimum energies in the steady state versus $\gamma_{max}$ decrease with increasing $\gamma_{max}$ below $\gamma_y$, but increase above, reaching the same values for $T = 1$ and $T = 0.466$. These data demonstrate the presence of a sharp transition between a low strain regime where oscillatory shear produces better annealed, hardened, glasses to a yielded regime displaying stress relaxation and rejuvenation.

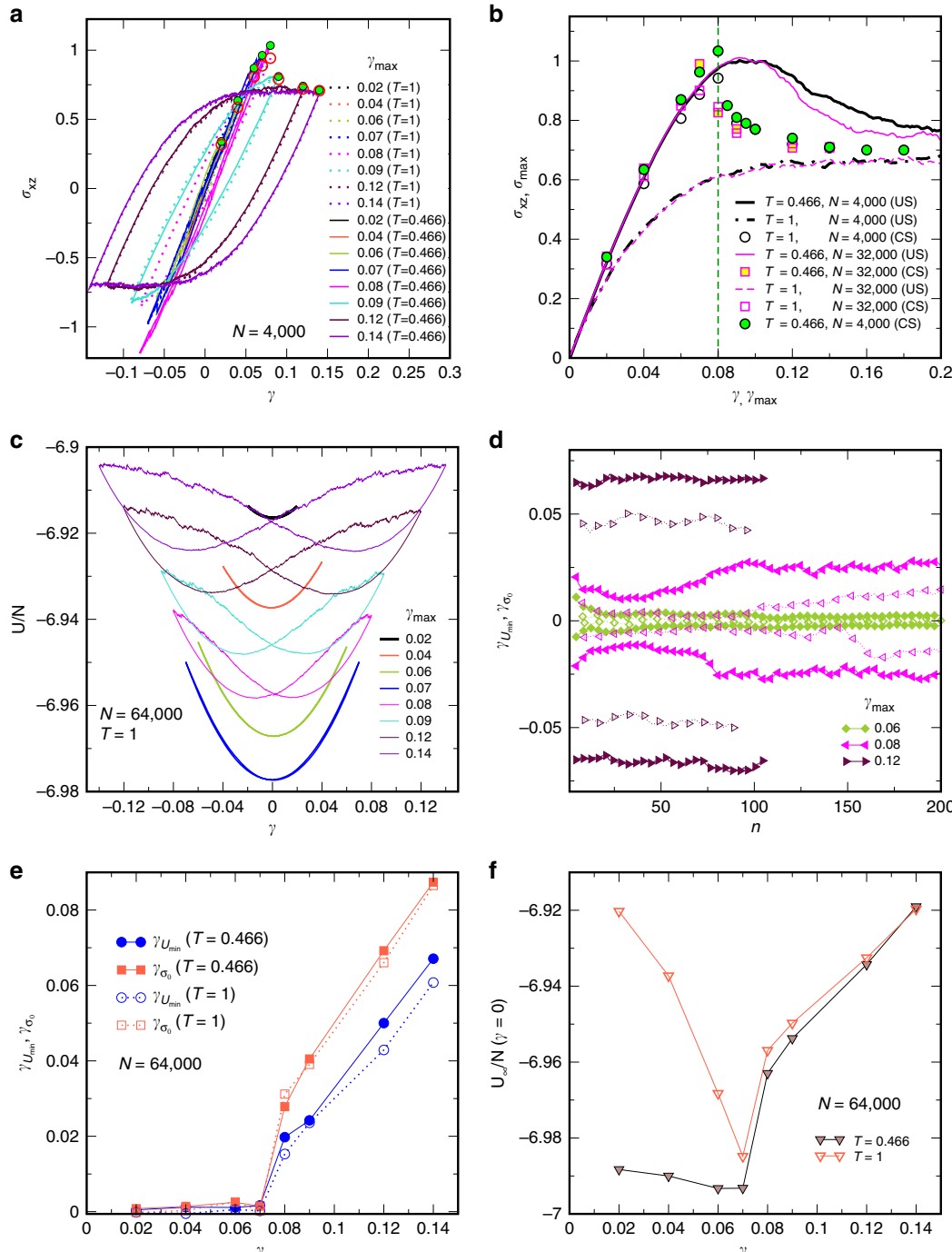

**Figure 1 | Stress and energy across the yielding transition. (a)** Stress–strain plots of the two differently annealed glasses for various strain amplitudes in the steady states of oscillatory shear deformation. Maximum stress in the cycle for each amplitude is marked by filled and open circles for $T = 0.466$ and $T = 1$, respectively. **(b)** Averaged stress–strain curves for uniform strain (US) are shown as lines—thick (black) for $N = 4,000$ and thin (magenta) for $N = 32,000$ while solid and dashed lines represent $T = 0.466$ and $T = 1$, respectively. Maximum stress $\sigma_{max}$ versus $\gamma_{max}$ are shown for cyclic strain (CS) (circle and square denote $N = 4,000$ and 32,000, respectively, with filled and open symbols corresponding to glasses from $T = 0.466$ and $T = 1$). The vertical line at $\gamma_{max} = 0.08$ indicates the sharp yielding transition seen. **(c)** Energy versus strain in the steady states, displaying a bifurcation in the strain corresponding to minima in energy at the yielding transition between $\gamma_{max} = 0.07$ and 0.08. **(d)** Strain values corresponding to energy minima ($\gamma_{U min}$) and and zero stress ($\gamma_{\sigma_0}$) are shown as open and filled symbols respectively, versus the number of cycles for different $\gamma_{max}$. For $\gamma_{max} = 0.08$ an initial relaxation towards zero is reversed as the system evolves to a yielded steady state with finite $\gamma_{U min}$ and $\gamma_{\sigma_0}$. **(e)** $\gamma_{U min}$ and $\gamma_{\sigma_0}$ as functions of strain amplitude $\gamma_{max}$, displaying a transition beyond $\gamma_{max} = 0.07$. **(f)** Asymptotic energy per particle at $\gamma = 0$ versus strain amplitude $\gamma_{max}$. Energies decrease with $\gamma_{max}$ until the yield strain is reached, after which they increase with $\gamma_{max}$.

**Statistics of avalanche sizes.** We next study (i) distribution of avalanche sizes, which we compute as the size of clusters of particles that undergo plastic rearrangements (see Methods for

how they are identified), and (ii) distributions of the size of energy drops. In Fig. 2a we show the distributions $P(s)$ of avalanche sizes $s$ for $N = 2,000$, which display a characteristic

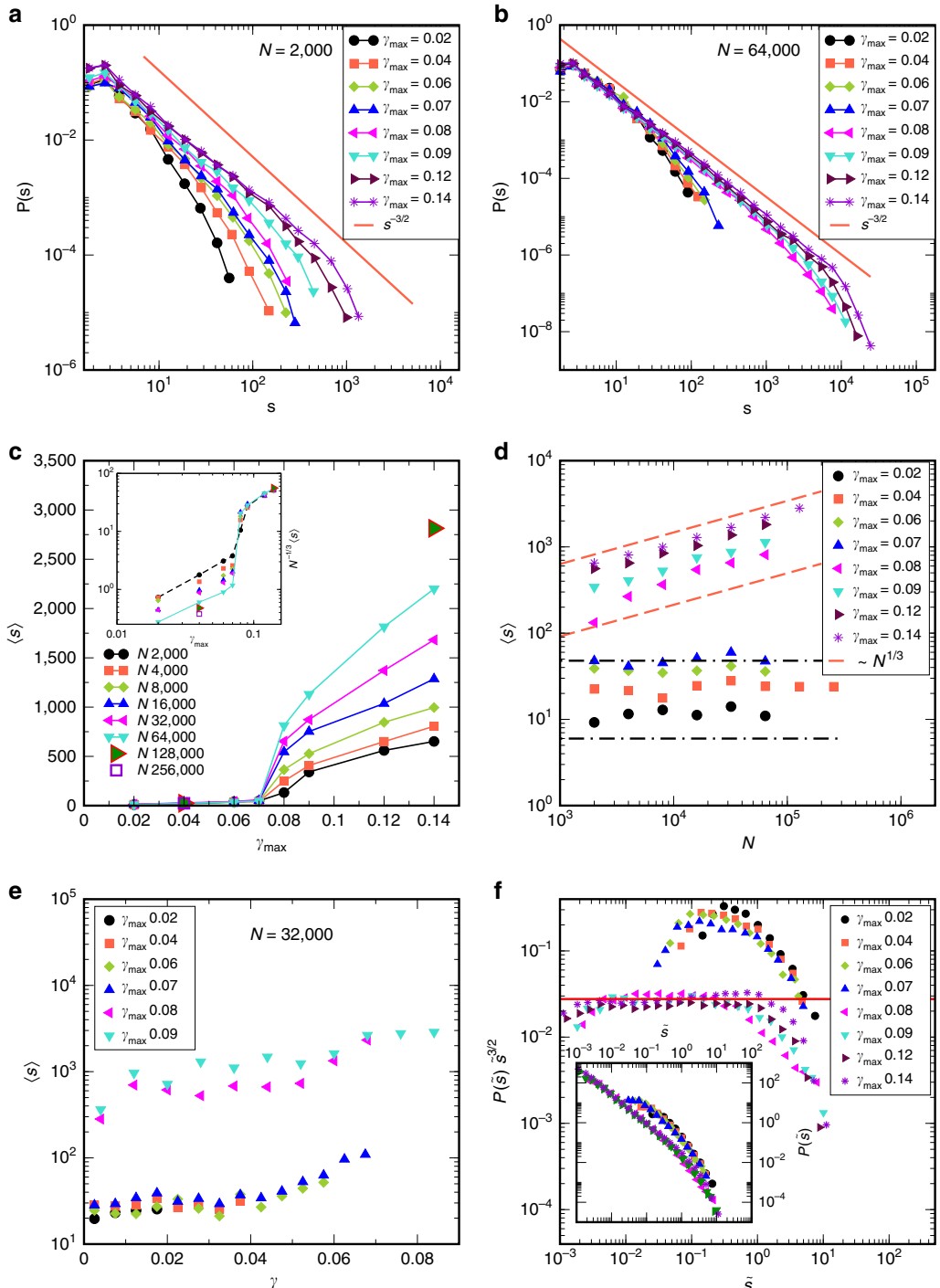

**Figure 2 | Statistics of avalanches as a function of strain amplitude $\gamma_{max}$ and system size $N$.** (**a**) Cluster size distributions for $N = 2,000$ displaying a power law with a cutoff that grows with $\gamma_{max}$ but does not indicate sharp changes at yielding. (**b**) Cluster size distribution for $N = 64,000$ displaying a sharp increase in the cutoff size across the yielding transition. The line in both panels corresponds to a power law with exponent $-3/2$. (**c**) Mean cluster size versus $\gamma_{max}$ showing a qualitative change across the yielding transition, with strong system size dependence above $\gamma_y$. The inset shows the mean cluster size scaled with $N^{1/3}$, which describes well the size dependence above $\gamma_y$. (**d**) Mean cluster size versus system size $N$ shows no significant size dependence for $\gamma_{max} < \gamma_y$ but a clear $N^{1/3}$ dependence above. A crossover in behaviour is seen for $\gamma_{max} = 0.08$. Lines, with $N^0$ (constant) and $N^{1/3}$ dependence, are guides to the eye. (**e**) Mean cluster sizes for bins in strain $\gamma$ for different $\gamma_{max}$ for $N = 32,000$. Mean cluster size does not depend on $\gamma_{max}$, and depends only mildly on strain $\gamma$, for two distinct sets, below and above yield strain $\gamma_y$. (**f**) Scaled cluster size ($\tilde{s} = s/\langle s \rangle$) distributions exhibit data collapse separately for $\gamma_{max} < \gamma_y$ and $\gamma_{max} > \gamma_y$ (inset). Distributions for $\gamma_{max} < \gamma_y$ do not display a power law regime, whereas $\gamma_{max} > \gamma_y$ do, over about two decades in $\tilde{s}$, as highlighted in a plot of $P(\tilde{s})\tilde{s}^{3/2}$ versus $\tilde{s}$. Data shown are for $T = 1$, and averages are over the full cycle, except for (**e**) which are averaged over the first quadrant.

power law decay with a cutoff. Although the cutoffs move to larger values as $\gamma_{max}$ increases, we see no indication of a transition. To assess the role of system sizes, we compute the avalanche sizes for a variety of system sizes. Figure 2b shows the avalanche size distribution for $N = 64,000$. The distributions fall into two clear sets, corresponding to $\gamma_{max}$ above and below $\gamma_y$. We compute and display in Fig. 2c the mean avalanche size $\langle s \rangle$ as a function of $\gamma_{max}$, for all studied system sizes. The striking observation is that below $\gamma_y$, $\langle s \rangle$ displays no system size dependence, and only a very mild dependence on $\gamma_{max}$, and no indication of the approach to $\gamma_y$. Above $\gamma_y$, a clear system size dependence is seen. Figure 2d shows the same data versus system size, revealing a roughly $N^{1/3}$ (or $\langle s \rangle \sim L$) dependence above $\gamma_y$, and minimal $N$ dependence below. The $N^{1/3}$ dependence is consistent with previous results[6,34] for mean energy drops, but the absence of system size dependence below, to our knowledge, has not been demonstrated before. We next ask whether the mean size of avalanches, for a given $\gamma_{max}$ depend on the strain $\gamma$ at which they appear, and conversely, for a given $\gamma$ what the dependence on $\gamma_{max}$ is. As shown in Fig. 2e ($N = 32,000$, $T = 1$), for a given $\gamma_{max}$ the $\gamma$ dependence is weak and is the same for $\gamma_{max} < \gamma_y$ (and $\gamma_{max} > \gamma_y$), but the data fall into distinct groups for $\gamma_{max} < \gamma_y$ and $\gamma_{max} > \gamma_y$. The same pattern is seen for the full distributions (Supplementary Fig. 5). For a given $\gamma_{max}$, the avalanche distributions can be collapsed on to a master curve by scaling $s$ by $\langle s \rangle$ (data not shown). The distributions of scaled sizes $\tilde{s} \equiv s/\langle s \rangle$, averaged over system size are shown in the inset of Fig. 2f. The same data are shown, multiplied by $\tilde{s}^{3/2}$ in the main panel, and demonstrate that the character of the distributions are different above and below yield: whereas above $\gamma_y$ one finds a range of sizes over which the power law form $P(s) \sim s^{-3/2}$ is clearly valid (and thus the cutoff arises purely because of system size), below $\gamma_y$ this is not the case, and the qualitative shape of the distributions is different (with a cutoff function multiplying the power law)[5,8,14,35].

We now discuss the distributions of energy drops. Shown for $N = 4,000$ and $32,000$ in Fig. 3a,b, these distributions show the same features as the avalanche sizes, but with a different power law exponent of $\sim 1.25$ (as found in ref. 15). Thus, the exponent depends on the quantity employed, and the avalanche size based on particle displacements is in closer agreement with mean field predictions. In Fig. 3c, we show the $\gamma_{max}$ dependence of the mean energy drop, for different system sizes, which reveal the same pattern as the avalanche sizes, albeit with a stronger apparent size dependence below yield. However, the total energy drops for the whole system include also an elastic component, in addition to the plastic component. The component of the energy drop corresponding to the plastic regions alone, which are plotted in Fig. 3d, to demonstrate that the plastic component has no system size dependence below yield. Figure 3e shows the system size dependence of the mean energy drop (plastic component), and Fig. 3f shows the mean energy drop versus $\gamma$ for different $\gamma_{max}$ ($N = 64,000$, $T = 1$), revealing the same separation below and above yield as the avalanche sizes. This is in contrast with the case of uniform shear, wherein both energy drops and avalanche sizes show a gradual, and strongly sample dependent, variation with strain (Supplementary Figs 6–9).

**Spatial structure of avalanches.** Finally, we analyse the spatial structure of the avalanches briefly, by studying (i) the percolation, and (ii) fractal dimension, of the avalanches. Below $\gamma_y$, none of the avalanches percolate, whereas above, a finite fraction does so. Figure 4a shows the weight of the spanning cluster $P_\infty$, and percolation probability $PP$ averaged over bins in 'probability' $P$, obtained from the fraction of displaced particles, (see Methods) for different system sizes for $\gamma_{max} = 0.08$, indicating a percolation

transition at $P \gtrsim 0.05$. However, the threshold is system size dependent, and thus merits further investigation. In Fig. 4b, $P_\infty$ and $PP$ averaged over all considered events are shown as a function of $\gamma_{max}$. The percolation probability does not become 1, a result of considering all the drop events. To address this artefact we analyse the cumulative set of all particles displaced in any of the events. The $P_\infty$ and $PP$ values shown in Fig. 4c indicate that above $\gamma_y$, this cumulative set always percolates and the weight $P_\infty$ is comparable for different system sizes. However, $P_\infty$ at the smallest $\gamma_{max}$ above $\gamma_y$ appears to increase with system size, suggesting a discontinuous change across $\gamma_y$. The variation of $P$ with $\gamma_{max}$ in either method also shows an apparently discontinuous behaviour across $\gamma_y$ (Supplementary Fig. 10).

We compute the fractal dimension of the spanning clusters using the box counting method (see Methods). Figure 4d shows a log-log plot of the occupied boxes versus magnification $r$ (the largest $r$ corresponds to the smallest box size, of $1.1\sigma_{AA}$) for $\gamma_{max} = 0.08$, $N = 32,000$. We find a fractal dimension of $d_f = 2.05$, close to 2, which appears consistent with the possibility that yield events are quasi-two dimensional. We find a fractal dimension of $d_f = 2.05$, close to 2, which is in consistent with the appearance of shear bands above the transition which are quasi-two dimensional (Supplementary Fig. 11). However, based on the system size dependence of the mean cluster size, the fractal dimension deduced is $d_f \sim 1$ (ref. 15), which is at odds with the result here, and requires further investigation for it to be properly understood.

## Discussion

The results that we have discussed demonstrate that a sharp yielding transition is revealed through oscillatory deformation of model glasses. The character of the avalanches is qualitatively different across the transition, being localized below the transition, and becoming extended above. Contrary to theoretical expectations for uniform deformation, the mean size of the avalanches does not diverge upon approaching the yielding transition, and prompts theoretical investigation, including development of suitable elasto-plastic models, of yielding under oscillatory deformation[36]. A signature of yielding is instead revealed by the progressive sluggishness of annealing behaviour as the transition is approached. Both the avalanche statistics and percolation characteristics suggest a discontinuous yielding transition, which may be consistent with the suggestion that yielding is a first order transition[12,16,37], but a comprehensive characterization of the nature of the transition requires further investigation. Finally, our results reveal systematic, non-trivial annealing behaviour of the glasses near the yielding transition, which we believe are of relevance to thermomechanical processing of metallic glasses. In particular, processing near the yielding transition, both above and below, may lead to significant change of properties, which may be utilized according to specific design goals.

## Methods

**Model.** The model system we study is the Kob-Andersen binary (80:20) mixtures of Lennard Jones particles. The interaction potential is truncated at a cutoff distance of $r_{c\alpha\beta} = 2.5\sigma_{\alpha\beta}$ such that both the potential and the force smoothly go to zero as given by

$$V_{\alpha\beta}(r) = 4\epsilon_{\alpha\beta}\left[\left(\frac{\sigma_{\alpha\beta}}{r}\right)^{12} - \left(\frac{\sigma_{\alpha\beta}}{r}\right)^6\right] + 4\epsilon_{\alpha\beta}\left[c_{0\alpha\beta} + c_{2\alpha\beta}\left(\frac{r}{\sigma_{\alpha\beta}}\right)^2\right], r_{\alpha\beta} < r_{c\alpha\beta} \quad (1)$$

where $\alpha, \beta \in \{A, B\}$ and the parameters $\epsilon_{AB}/\epsilon_{AA} = 1.5$, $\epsilon_{BB}/\epsilon_{AA} = 0.5$, $\sigma_{AB}/\sigma_{AA} = 0.80$, $\sigma_{BB}/\sigma_{AA} = 0.88$. Energy and length are in the units of $\epsilon_{AA}$ and $\sigma_{AA}$, respectively, and likewise, reduced units are used for other quantities. The correction terms $c_{0\alpha\beta}$, $c_{2\alpha\beta}$ are evaluated with the conditions that the potential and its derivative at $r_{c\alpha\beta}$ must vanish at the cutoff.

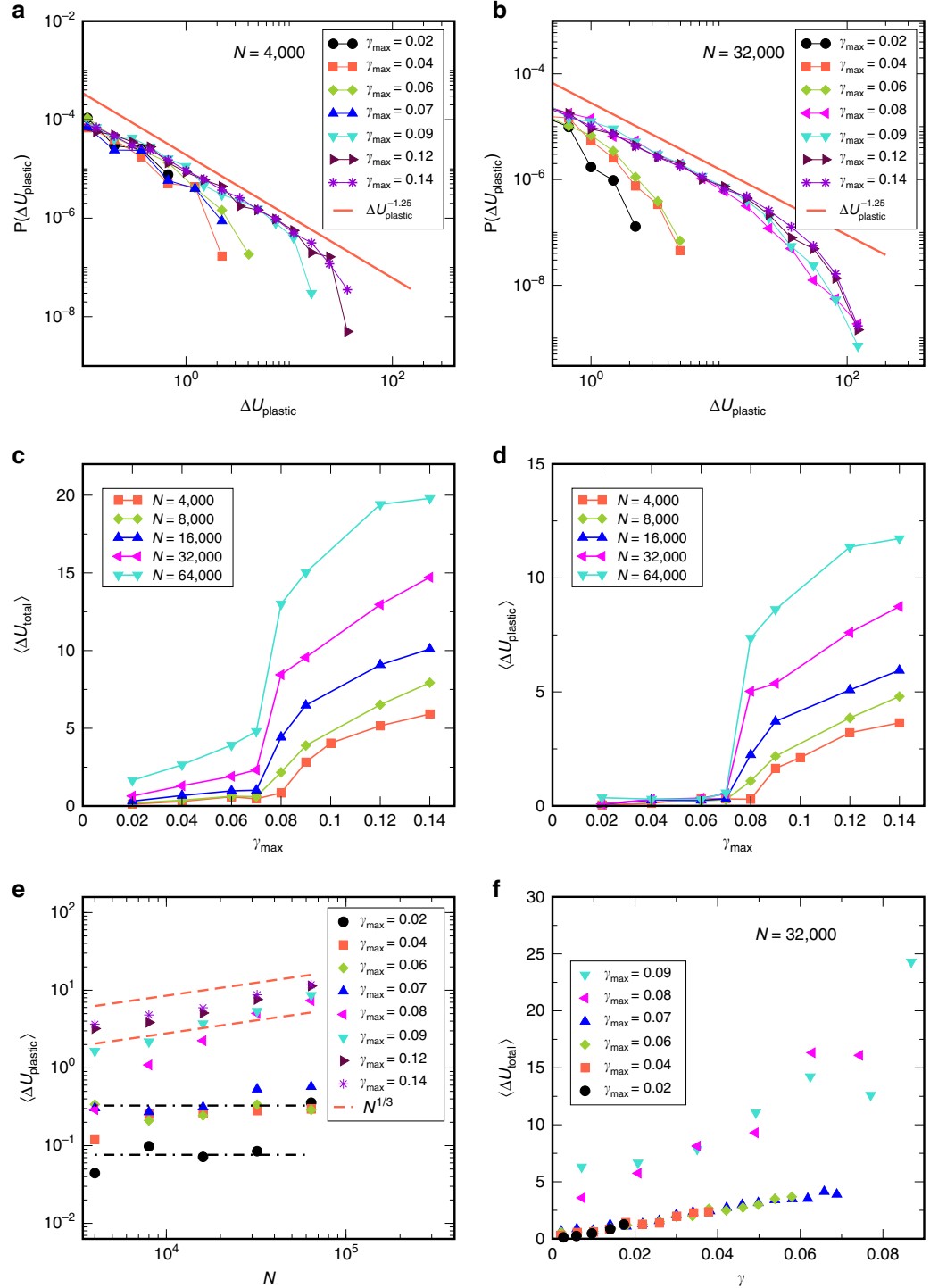

**Figure 3 | Statistics of energy drops as a function of strain amplitude $\gamma_{max}$ and system size N.** Distributions of energy drops (**a**) for $N = 4,000$ show no clear separation of $\gamma_{max} < \gamma_y$ and $\gamma_{max} > \gamma_y$, whereas (**b**) for $N = 32,000$ a clear separation is visible. In both cases, a power law regime is apparent, with exponent $\sim 1.25$. (**c**) Mean energy drops versus $\gamma_{max}$, indicating a sharp change at $\gamma_y$. (**d**) Mean energy drops considering only plastic regions show no system size dependence below $\gamma_y$. (**e**) Mean energy drop (plastic component) versus system size $N$ shows no significant size dependence for $\gamma_{max} < \gamma_y$ but a clear $N^{1/3}$ dependence above. A crossover in behaviour is seen for $\gamma_{max} = 0.08$. Lines, with $N^0$ (constant) and $N^{1/3}$ dependence, are guides to the eye. (**f**) Mean energy drops (total) for bins in strain $\gamma$ for different $\gamma_{max}$ for $N = 32,000$, $T = 1$ showing no dependence on $\gamma_{max}$, and only a mild dependence on strain $\gamma$, for two distinct sets, below and above yield strain $\gamma_y$. Data shown are for $T = 1$, and averages are over the first quadrant.

**Initial glass configurations.** The initial liquid samples are equilibrated at two temperatures, $T = 0.466$ and $T = 1$ using the Nosé Hoover thermostat, at reduced density $\rho = 1.2$. Independent samples are generated for each temperature and system size by further evolving the equilibrated liquid configurations by performing the molecular dynamics simulations of constant energy, which are separated by the structural relaxation time ($\tau_\alpha$) obtained from the self intermediate scattering function ($F_s(k, t)$). For the uniform shearing data, we have atleast 100 samples for all the system sizes. The avalanche data shown for cyclic shearing are for at least 20

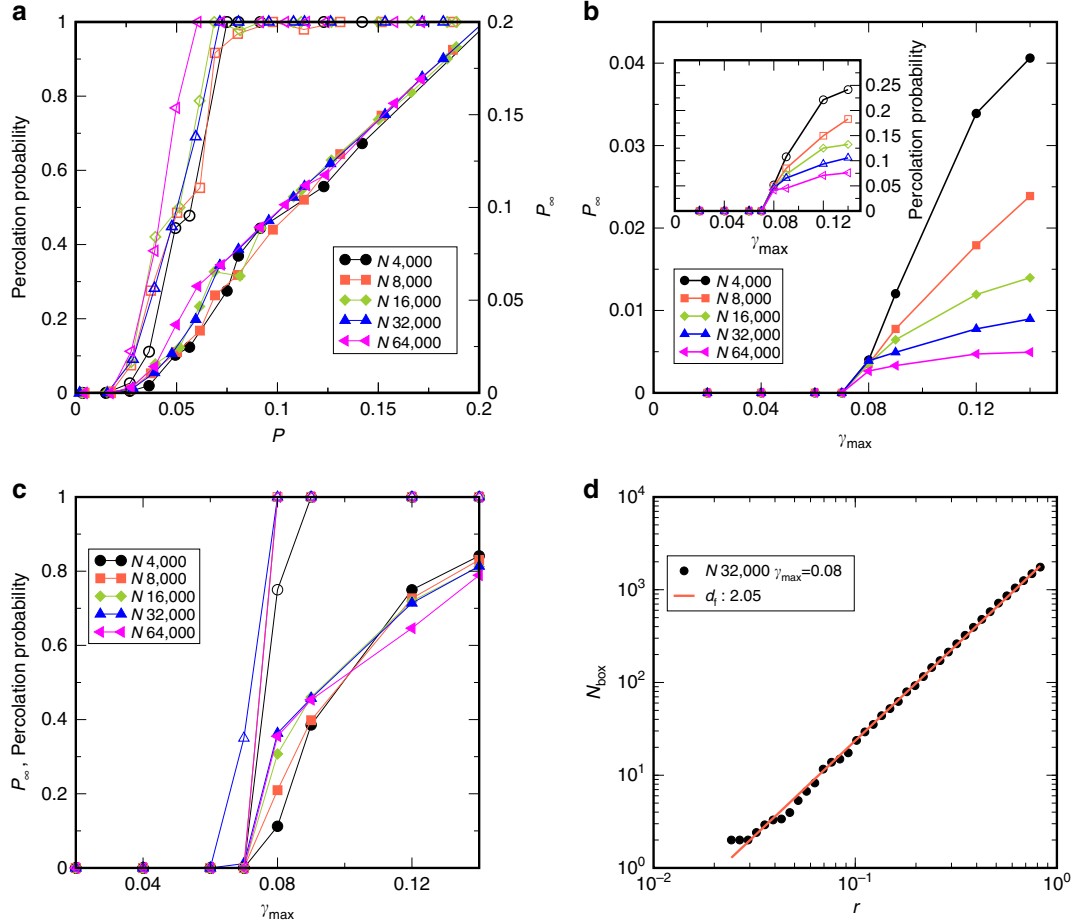

**Figure 4 | Percolation of avalanches and fractal dimension of percolating clusters. (a)** Percolation probability and weight of the spanning cluster $P_\infty$ shown as open and filled symbols, respectively, against the occupation number $P$ for different system sizes, considering all events, for $\gamma_{max} = 0.08$. A percolation transition takes place for $P \simeq 0.05$ although the threshold is system size dependent. **(b)** Percolation probability (inset) and $P_\infty$ averaged over all events, versus $\gamma_{max}$. **(c)** Percolation probability and $P_\infty$ for the cumulative set of particles rearranging over a cycle, shown as open and filled symbols respectively versus $\gamma_{max}$, indicating a percolation transition at the yielding strain $\gamma_y$. $P_\infty$ just above the transition increases with system size. **(d)** Fractal dimension estimation from box counting. A log-log plot of the number of occupied boxes ($N_{box}$) is shown versus the magnification $r$. The slope results in an estimated fractal dimension $d_f = 2.05$. Data shown are for $T = 1$, and averages are over the first quadrant.

samples for $N \leq 32,000$, and 10 samples for larger systems. All the simulations are carried out using LAMMPS[38].

**Simulation details.** Shear deformation of the model amorphous solids is done employing athermal-quasi static (AQS) simulations which consist of two steps. An affine transformation of coordinates $x' = x + d\gamma \times z$; $y' = y$; $z' = z$ is imposed, subsequently followed by an energy minimization using the conjugate-gradient method with Lees-Edwards periodic boundary conditions. Strain steps of $d\gamma = 2 \times 10^{-4}$ are used throughout, except for $N = 256,000$ for which $d\gamma = 5 \times 10^{-4}$. Initial configurations are the inherent structures (local energy minima) of equilibrated liquid samples. Potential energy and mean square displacements are computed at $\gamma = 0$ as functions of cycles to ascertain that steady states are reached, wherein the coordinates of particles, and properties such as the potential energy $U$ and shear stress $\sigma_{xz}$ remain (below yield strain) unchanged at the end of each cycle, or (above yield strain) become statistically unchanged upon straining further, and exhibit diffusive motion as a function of the number of cycles. Steady states for strain amplitudes of $\gamma_{max} = 0.02, 0.04, 0.06, 0.07, 0.08\ 0.09, 0.12, 0.14$ are studied for system sizes $N = 2,000, 4,000, 8,000, 16,000, 32,000$ and $64,000$. To further probe finite size effects, we have consider amplitude below the yield transition at $\gamma_{max} = 0.04$ for $N = 128,000$ and $256,000$ and $\gamma_{max} = 0.14$ for $N = 128,000$.

**Identifying avalanches.** In the steady state, we compute the potential energy per particle and stress for each strain step. Plastic events result in discontinuous energy and stress drops. A parameter $\kappa = \frac{\delta U}{Nd\gamma^2}$(ref. 34) exceeding a value of 100 is used to identify plastic events, where $\delta U$ is the change in energy during minimization after a strain step. Avalanche sizes based on the magnitude of energy drops and the cluster sizes of 'active' particles (that undergo plastic displacements) are both computed. Particles are considered active if they are displaced by more than

$0.1\sigma_{AA}$. The choice of this cutoff is based on considering the distribution of single particle displacements $\delta r$, which are expected to vary as a power law $P(\delta r) \sim \delta r^{-5/2}$ for elastic displacements around a plastic core, but display an exponential tail corresponding to plastic rearrangements (see, for example, ref. 30). The separation is clear cut only for small $\gamma_{max}$, and we choose the smallest cutoff value (observed for $\gamma_{max} = 0.02$) so that plastic rearrangements at all $\gamma_{max}$ are considered. In performing cluster analysis, two active particles are considered to belong to the same cluster if they are separated by $< 1.4\sigma_{AA}$ (first coordination shell). The normalized histogram of cluster sizes $P(s)$ is obtained from statistics for all the events. The mean cluster size is computed from the distributions as $\langle s \rangle = \frac{\sum_s s^2 P(s)}{\sum_s s P(s)}$ (Supplementary Fig. 4).

**Percolation analysis.** For the percolation analysis, we consider all the plastic events in the first quadrant of the cycle ($\gamma$ from 0 to $\gamma_{max}$), and compute the 'probability' $P$ from the fraction of particles that undergo plastic displacement, and the weight of the spanning cluster $P_\infty$, from the fraction of particles that belong to the spanning cluster ($P_\infty = 0$ if there is no spanning cluster). The percolation probability $PP = 1$ if a spanning cluster is present and 0 otherwise.

**Fractal dimension.** To obtain the fractal dimension of percolating clusters, we employ the method of box counting. The simulation volume is divided into boxes of a specified mesh size, and the number of boxes that contain a part of the cluster, $N_{box}$, is counted. This is repeated for a series of mesh sizes, and the fractal dimension is obtained as the slope $d_f = \frac{\log(N_{box})}{\log r}$ where $r$ is the inverse of mesh size.

**Data availability.** The data that support the findings of this study are available from the corresponding author upon request.

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

## Acknowledgements

We wish to thank J.L. Barrat, P. Chaudhuri, M. Falk, G. Foffi, A.L. Greer, J. Horbach, I. Procaccia, M. Robbins, A. Rosso and M. Wyart for useful discussions. We wish to specially thank H.A. Vinutha for discussions and help regarding computations reported here. We gratefully acknowledge TUE-CMS and SSL, JNCASR, Bengaluru for computational resources and support.

## Author contributions

S.S. conceived the project and supervised the research. P.L. and A.D.S.P. performed the simulations and data analysis. P.L., A.D.S.P. and S.S. analysed and interpreted the results, and wrote the paper.

## Additional information

**Competing financial interests:** The authors declare no competing financial interests.

**Publisher's note**: 

