## [Peer Review File · Nature Communications]

PEER REVIEW FILE

Reviewers' Comments:

Reviewer #1 (Remarks to the Author):

The authors consider a generic model of a binary glass and study the yielding transition under oscillatory shear deformation using molecular dynamics simulations. This is an important topic that has recently received considerable attention including ref 12 in Nature Communications and it will be of interest for a broad community of researchers who study colloidal glasses, metallic glasses and other disordered systems under active deformation. In the present study, extensive computer simulations are performed in a wide range of system sizes, strain amplitudes, preparation histories to probe the yield point via oscillatory shear. In particular it is demonstrated that the mean size of avalanches does not diverge upon approaching the yielding transition from below, but it grows above the yield point. The statistics of percolation clusters suggests a discontinuous yielding transition, that might indicate that yielding is a first order transition. I recommend publication of the manuscript in Nature Communications after the following comments are addressed.

- 1) In the analysis of the avalanche statistics and percolation, do the clusters form shear bands at large strain amplitudes (similar to what was reported in ref 23 for a large system)? Is this related to the fact that the fractal dimension of these clusters is about 2? Is there a certain thickness of shear bands and how does it depend on the strain amplitude?
- 2) Are the avalanches in steady state near yield reversible? How does it affect the avalanche statistics and the mean cluster size? If the clusters are reversible then a large number of independent samples are required.
- 3) How does Fig.2 in Supplementary Information look for well annealed glasses?

Reviewer #2 (Remarks to the Author):

The paper by Leishangthem et al. investigates the “yielding” of glasses under oscillatory shear using an athermal quasistatic simulation protocol. The authors report a sharp transition at a critical strain $\gamma_y \sim 0.07$, accompanied by a distinct change of character of avalanches from localized to system spanning. These results are interesting, timely, and well evaluated.

Furthermore, the simulations seem well performed and of high quality. The interpretation in terms of a sharp transition, and the analogy with an equilibrium transition, though not entirely new and observed before, are in my view justified and well done.

I have, however, several concerns related to the context of the work, and confusion related to the use of the term “yielding”. The authors refer mostly only to simulation work on the straining of glasses, while much experimental work has been done recently. Citing only simulation work makes the paper marginal and not suited for a high-impact journal with a broad scope such as Nature Communications. At the end, it is the combination of experiments, simulations and modelling that lead to complete understanding. In particular, there are several experimental works that directly observe signatures of a sharp transition under oscillatory strain, most notably the work by Denisov et al., “Sharp symmetry-change marks the mechanical failure transition of glasses” Scientific Reports 5, 14359 (2015), where precisely this sharp “yielding” transition of a glass to oscillatory strain has been observed. Obviously, this must be cited in this context to make the claims of the authors much more substantial and relevant.

Second, the term “yielding” is very much confusing here. As the authors surely know, there has been much confusion in the field as to whether the “yielding” transition is a continuous or a sharp transition. Much of this confusion stems from a very loose use of the word “yielding”, which is traditionally used for the elasto-plastic transition of a material in constant strain-rate experiments, and should be only used in this context. The flow of a material under oscillatory strain (in rheology referred to as transition from linear to nonlinear regime) is a completely different scenario that should not be confused with constant strain rate experiments! In this sense explicitly referring to and comparing with avalanches in the const. strain rate yielding is misleading. Why should they be the same? This is a very different protocol with likely very different steady states that should be clearly distinguished. The author should clearly make this distinction by avoiding the word “yielding” wherever possible, and by very clearly stating the difference to the usual const. strain rate yielding.

More details are given below. I believe this paper has good potential and may eventually be publishable in Nat. Comm., but the authors should carefully address all points in a revised version.

1. The introduction refers mainly (almost exclusively) to simulation work; experimental work must be cited here as well, in particular the observation of a sharp transition under oscillatory shear as mentioned above. (e.g. “Most analyses have employed computer simulations of atomistic models of glasses, aiming to elucidate key features of plastic response”. No, there are

very relevant experiments as well, especially those with single-particle tracking. “However, barring some recent work [], it is not been employed widely to probe yielding in amorphous solids computationally” Again misleading: experimentally highly relevant evidence has already been published!)

2. Don't use the word yielding in another context. Referring to “.. a theoretical description of yielding in amorphous solids predicts the mean avalanche size to diverge as the yielding transition is approached ...” is very confusing. The author's oscillatory protocol is very different, and this should be clearly stated! “Here we show that oscillatory deformation offers a robust approach to systematically probe behaviour above and below yielding...” Again confusing: no, this is NOT the same type of yielding, please don't use the word yielding and clearly distinguish!

3. It is not clear in the main manuscript how the athermal quasistatic shear is performed in oscillatory straining. Is the system relaxed many times on the way to γ_{max} , or just two times in the cycle (at γ_{max} and $-\gamma_{max}$). I know this is explained in the SI and becomes clear later, but it should be clearly described on p. 3 already.

4. On a related note, this AQS protocol is less known, and it is not clear at all how it compares to real experimental situations. For the more known uniform quasistatic shearing protocol, this is established, but for the oscillatory equivalent this is much less known, and in my opinion much less obvious. The authors should at least create some confidence by stating how this protocol describes reality.

5. The interpretation of a sharp transition in Fig. 1 is nice; still, this is a nonequilibrium transition. Can the authors use their data to elucidate the nonequilibrium nature of the transition? E.g. dissipation is a central property of nonequilibrium, with the dissipated energy reflected in the area of the stress-strain loop in Fig. 1a. How is this related to the “Landau-like” bimodal energy landscape? How do these two “steady-state” minima arise? (Probably combination of elastic and plastic straining in the oscillatory cycle?)

6. In fact, there is a series of experiments by Rogers et al., J. Rheology (e.g. J. Rheol. 55, 435 (2011)) that apply precisely the same analysis of plotting σ_{max} values from the Lissajous figures as done in Fig. 1a and b. How do the current simulations compare to these measurements. Was a similarly sharp drop of σ_{max} as that in Fig. 1b observed already in these experiments? The authors should refer, at least briefly, to these measurements.

7. The authors talk about “sluggishness” of the response upon approaching γ_y . When they mention it on page 3, it is unclear what they actually mean. I suggest rephrasing this to clarify the meaning. Later in Fig. 1d it becomes clear that the system close to γ_y needs

some time (cycles) to decide where to go; while this makes intuitive sense, what does it actually mean physically? In an equilibrium transition, nucleation plays a central role in delaying the onset of the transition. Is there some similar “nucleation” happening here in the nonequilibrium case? What does the spatial distribution look like; is it localised?

8. While I generally believe the author’s interpretation of a sharp transition, some of the graphs do not look like sharp step functions. E.g. if I look at Fig. 1e this could as well be a continuous transition (critical point) entering a “two phase regime”. This is also true for the avalanche sizes in Fig. 2c. This certainly can have to do with the still limited system size, and the increasing sharpness with system size in Fig. 2c seems to point in this direction; still, there is no real proof that the transition itself is sharp. I agree that the qualitative change in behaviour points towards a “coexistence” regime; but couldn’t this be entered through a critical point? In this sense, I would have liked to see the data points for $N > 64000$ that I see in the legend, but unfortunately not in the relevant regime in the graph in Fig. 2c. Maybe they are hidden behind the inset? As this is such a central point of debate in the community right now, it would be valuable to know how sharp the transition really is.

9. Citing only theoretical work on p. 6 is misleading (“... which may be consistent with the suggestion that yielding is a first-order transition [14].”) There is earlier experimental work reporting a sharp “yielding” transition in oscillatory rheology, see in particular Denisov et al., Scientific Reports 5, 14359 (2015). This should be cited here, as it is most relevant and indeed in excellent agreement with the authors’ results.

Reviewer #3 (Remarks to the Author):

A. In this paper the authors study numerically the on-set of the yielding transition using a novel technique: the oscillatory shear deformation. Their main result is the presence of a sharp transition with size dependent avalanches above threshold, and localised avalanches below.

B. the technique is novel and allows to target the steady state even below threshold where most of the studies focus on a transient regime

C-D. the method and the quality of the data appear at the state of the art of the domain

E. I am totally convinced about the existence of the transition and the presence of size dependent avalanches above threshold. I am still surprised about the fact that the mean size of the avalanches does not diverge upon approaching the yielding transition. Is this divergence hidden by the novel protocol that mixes avalanches at different stress values? In the future further investigation on this point are needed

F. The paper is well written and self-contained. I think it deserves publication in Nature Communication

G. The references are appropriate

H. The abstract, the introduction and the conclusion are very clear

Response to Referee Comments

NCOMMS-16-16710-T

“The yielding transition in amorphous solids under oscillatory shear deformation”

Premkumar Leishangthem, Anshul D. S. Parmar, Srikanth Sastry

We thank all the referees for reviewing our manuscript, and their positive assessment of our manuscript. We have made an effort to address the questions in the responses below, and in the revised manuscript. We hope that the referees find the responses satisfactory and find our manuscript suitable for publication.

In what follows, we have reproduced the referees' comments (in full) in black/normal text, our responses blue text, and changes to the manuscript are quoted in red/boldface text.

Referee-1 Comments

The authors consider a generic model of a binary glass and study the yielding transition under oscillatory shear deformation using molecular dynamics simulations. This is an important topic that has recently received considerable attention including ref 12 in Nature Communications and it will be of interest for a broad community of researches who study colloidal glasses, metallic glasses and other disordered systems under active deformation. In the present study, extensive computer simulations are performed in a wide range of system sizes, strain amplitudes, preparation histories to probe the yield point via oscillatory shear. In particular it is demonstrated that the mean size of avalanches does not diverge upon approaching the yielding transition from below, but it grows above the yield point. The statistics of percolation clusters suggests a discontinuous yielding transition, that might indicate that yielding is a first order transition. I recommend publication of the manuscript in Nature Communications after the following comments are addressed.

We thank the referee for the positive recommendation. Our responses are given point-wise below.

Q1. In the analysis of the avalanche statistics and percolation, do the clusters form shear bands at large strain amplitudes (similar to what was reported in ref 23 for a large system)? Is this related to the fact that the fractal dimension of these clusters is about 2? Is there a certain thickness of shear bands and how does it depend on the strain amplitude?

[REDACTED]

[Redacted text block]

[Redacted text block]

[Redacted text block]

[Redacted text block]

Q2. Are the avalanches in steady state near yield reversible? How does it affect the avalanche statistics and the mean cluster size? If the clusters are reversible then a large number of independent samples are required.

As we understand the referee’s question, the reversibility of avalanches refers to a return to the initial configuration at the end of each cycle.

To ensure clarity of what is meant, we distinguish reversibility of two kinds. (1) Reversibility with respect to strain step. In this case, for each strain increment, when we reverse the strain direction the particles come back to the same position. (2) Reversibility of configuration at the end of each cycle. Despite undergoing many plastic jumps within a cycle, at the completion of a cycle of deformation the positions of particles return to the positions at the start of the cycle. This has been referred to in related literature as “loop reversibility”.

For the system we study, the avalanches are not reversible on either side of the transition in the first sense. If the avalanches were reversible with respect to strain direction one would observe the loop area (in a stress-strain plot) to be zero at some strain amplitudes. However, we find that there is a small but finite value of loop area even below the transition, as shown in Fig. 3.

Now, in the second sense, we have loop reversibility below the yielding transition. Thus in the steady state, cyclically sheared configurations return to their initial state at the end of the cycle, and thus no independent cycling is possible by further cycles of deformation. Instead, we must sample independent initial configurations. We have used at least 20 independent samples for all strain amplitudes. Here, for each sample, we consider all the plastic events occurring in a cycle of the steady state. Above the yield transition, there is no loop reversibility, and hence we may treat different cycles in the steady state as independent samples.

Figure 3: Hysteresis loop area *vs.* strain amplitude (linear-linear and log-log scales) for $N = 32000, 64000$ of $T = 1$.

Q3. How does Fig.2 in Supplementary Information look for well annealed glasses?

Below the transition strain amplitude, both well and poorly annealed glasses display decreasing energies with strain amplitude. But, above the transition, the poorly annealed glass shows a decrease in energy with cycles whereas the well annealed glass shows increasing energies with cycles. The comparison of the behaviour for the two temperatures we study are shown below. The data for the well annealed glasses is now included in our Supplementary Information.

Figure 4: Energy vs. strain for $N = 64000$, $T = 0.466$ with $\gamma_{max} = 0.06$ and 0.12 .

Figure 5: Energy vs. strain for $N = 64000$, $T = 1$ with $\gamma_{max} = 0.06$ and 0.12 .

In Fig. 2 of the Supplementary Information, we now show the data for well annealed glasses in addition to the poorly annealed glasses, to demonstrate the difference between the two during their evolution towards the steady states.

Referee-2 Comments

The paper by Leishangthem et al. investigates the “yielding” of glasses under oscillatory shear using an athermal quasistatic simulation protocol. The authors report a sharp transition at a critical strain $\gamma_y \sim 0.07$, accompanied by a distinct change of character of avalanches from localized to system spanning. These results are interesting, timely, and well evaluated. Furthermore, the simulations seem well performed and of high quality. The interpretation in terms of a sharp transition, and the analogy with an equilibrium transition, though not entirely new and observed before, are in my view justified and well done. I have, however, several concerns related to the context of the work, and confusion related to the use of the term “yielding”. The authors refer mostly only to simulation work on the straining of glasses, while much experimental work has been done recently. Citing only simulation work makes the paper marginal and not suited for a high-impact journal with a broad scope such as Nature Communications. At the end, it is the combination of experiments, simulations and modelling that lead to complete understanding. In particular, there are several experimental works that directly observe signatures of a sharp transition under oscillatory strain, most notably the work by Denisov et al., “Sharp symmetry-change marks the mechanical failure transition of glasses” Scientific Reports 5, 14359 (2015), where precisely this sharp “yielding” transition of a glass to oscillatory strain has been observed. Obviously, this must be cited in this context to make the claims of the authors much more substantial and relevant. Second, the term “yielding” is very much confusing here. As the authors surely know, there has been much confusion in the field as to whether the “yielding” transition is a continuous or a sharp transition. Much of this confusion stems from a very loose use of the word “yielding”, which is traditionally used for the elasto-plastic transition of a material in constant strain-rate experiments, and should be only used in this context. The flow of a material under oscillatory strain (in rheology referred to as transition from linear to nonlinear regime) is a completely different scenario that should not be confused with constant strain rate experiments! In this sense explicitly referring to and comparing with avalanches in the const. strain rate yielding is misleading. Why should they be the same? This is a very different protocol with likely very different steady states that should be clearly distinguished. The author should clearly make this distinction by avoiding the word “yielding” wherever possible, and by very clearly stating the difference to the usual const. strain rate yielding.

More details are given below. I believe this paper has good potential and may eventually be publishable in Nat. Comm., but the authors should carefully address all points in a revised version.

We very much appreciate the referee’s point of view regarding the combined application of experimental, theoretical and computational approaches to generate understanding, and fully agree with the referee. We did have a few references to experimental work, but agree that our referencing was biased towards simulation studies. In large part, this arose from the restrictions on the maximum number of citations in an earlier submission of this work, and the necessity to make specific references to simulation work for direct comparison with our own. We have remedied this drawback in our revised manuscript and thank the referee for alerting us to the danger that our manuscript risked being viewed as having limited relevance to other simulation studies. We address specific remarks of the referee below.

Q1. The introduction refers mainly (almost exclusively) to simulation work; experimental work must be cited here as well, in particular the observation of a sharp transition under oscillatory shear as mentioned above. (e.g. “Most analyses have employed computer simulations of atomistic models of glasses, aiming to elucidate key features of plastic response”. No, there

are very relevant experiments as well, especially those with single-particle tracking. “However, barring some recent work [], it is not been employed widely to probe yielding in amorphous solids computationally” Again misleading: experimentally highly relevant evidence has already been published!)

As stated above, we appreciate very much the referee’s warning about a biased referencing of previous work. However, in the quoted text, our intended meaning was different. Firstly, we simply wanted to state that many (not most) analyses have **also** used computer simulations to elucidate plastic response. This was intended simply as a background statement to our own work, and not an exclusion of work using other approaches. Secondly, we meant to state that among computational studies, oscillatory deformation has not been employed widely to probe yielding. Again, this was not meant as a comment on the absence of experimental studies. However, since our text and citations conveyed a different meaning than we intended, we have made the following changes to be more accurate and clear:

1. In the revised manuscript, in addition to references to experimental work we had included earlier (Refs. 7, 9, 10), we now have added several references to experimental work addressing plastic response under large amplitude oscillatory shear deformations. In particular, we are happy to cite the work of Denisov *et al.*, “Sharp symmetry-change marks the mechanical failure transition of glasses” *Scientific Reports* 5, 14359 (2015), since the anisotropy of structure – addressed by Denisov *et al* – is something we are attempting at present to understand better, in part motivated by their results.
2. We have modified the text in several places to make our meaning more transparent, as listed below:
 1. We now cite seven additional references of experimental work on yielding transition using the oscillatory shear deformation. These are cited as Ref. 12 in the opening paragraph, and on page 3 as [24-29]. References have been slightly reordered as a result.
 2. We have modified the line
“Many analyses have employed computer simulations of atomistic models of glasses, aiming to elucidate key features of plastic response [] on atomic scales.”
to
“In addition to extensive experimental and theoretical investigations, computer simulations of atomistic models of glasses have also been employed, to elucidate key features of plastic response [] on atomic scales.” to more accurately reflect our intended meaning.
3. We have modified the text
“However, barring some recent work [], it is not been employed widely to probe yielding in amorphous solids computationally.”
to
“However, it has not been employed widely in computational investigations, barring some recent work [], to probe yielding in amorphous solids.”

Q2. Don’t use the word yielding in another context. Referring to “.. a theoretical description of yielding in amorphous solids predicts the mean avalanche size to diverge as the yielding transition is approached ..” is very confusing. The author’s oscillatory protocol is very different, and this should be clearly stated! “Here we show that oscillatory deformation offers a robust

approach to systematically probe behaviour above and below yielding...” Again confusing: no, this is NOT the same type of yielding, please don’t use the word yielding and clearly distinguish!

We agree with the referee’s insistence that we do not use the word “yielding” in a confusing manner, but would like to point out that our use of the term is consistent with the widely used meaning of the term in the relevant literature. Our current references 24-26 (Rogers. et al, Koumakis et al, Gibaud et al) use the word yielding in the experimental context of large amplitude oscillatory shear, and our current references 6 and 16 use the same terminology in the context of the computational AQS protocol. Thus, we would like to retain the use of this term, but have made an effort to ensure that no confusion is caused by alerting the readers in the contexts the referee mentions, as follows:

1. We have modified the sentence

“a theoretical description of yielding in amorphous solids [], predicts the mean avalanche size to diverge as the yielding transition is approached from below, leading to a power law distribution with a diverging mean size at and above the transition. “

to

“a theoretical description of mechanical failure in amorphous solids [], predicts the mean avalanche size to diverge as a critical stress is approached from below, leading to a power law distribution with a diverging mean size at and above the transition.”

which serves both to avoid the word yielding and to express the authors’ result being quoted in their own terminology.

2. We replace the statement

“Here, we show that oscillatory deformation offers a robust approach to systematically probe behaviour above and below yielding.”

with

“Here, we show that oscillatory deformation offers a robust approach to systematically probe behaviour above and below a sharply defined point of mechanical failure, which we associate with yielding. As our results pertain to oscillatory deformation in the limit of vanishing shear rate, we caution that comparisons with uniform shear at finite rates must be made with due care.”

which we believe addresses the referee’s concern about being careful in our use of terminology.

Q3. It is not clear in the main manuscript how the athermal quasistatic shear is performed in oscillatory straining. Is the system relaxed many times on the way to γ_{max} , or just two times in the cycle (at γ_{max} and $-\gamma_{max}$). I know this is explained in the SI and becomes clear later, but it should be clearly described on p. 3 already.

We thank the Referee for pointing out that our explanation of AQS was not self contained on page 3. The exact procedure used is given in the Methods section, but we have now added text to make clear the basic procedure.

We change in 2nd para of p. 3 from “ ... are subjected to volume preserving shear deformation through the AQS protocol.” to “... are subjected to volume preserving shear deformation through the AQS protocol, **wherein the strain γ_{xz} is incremented in small steps, with each step being followed by energy minimization.**”

Q4. On a related note, this AQS protocol is less known, and it is not clear at all how it compares to real experimental situations. For the more known uniform quasistatic shearing protocol, this is established, but for the oscillatory equivalent this is much less known, and in my opinion much less obvious. The authors should at least create some confidence by stating how this protocol describes reality.

The athermal quasistatic protocol represents a limit case where both the temperature and the shear rate are taken to zero. This limit is expected to be relevant in modelling the behaviour of glasses at very low temperatures compared to the glass transition temperature and in the limit where the relaxation processes in the material in response to applied stress take place fast compared to the rate of deformation. While the AQS limit needs thus to be properly contextualised, we believe that the case of oscillatory deformation using AQS does not need any additional justification. We understand that the referee wishes us to include a perspective to orient the reader and to indicate why one should have confidence in the results generated. We have done so as indicated below.

On page 2 we have added **“The AQS protocol represents a limit in which the deformation behaviour of the solids does not depend crucially on thermally induced processes, and relaxation processes are expected to occur on time scales much smaller than the inverse shear rate. Thus, results from AQS may be expected to be useful in understanding the behaviour of glasses sufficiently below the glass transition, and for small shear rates. Both these conditions may be expected to be satisfied in the context of understanding the mechanical failure of glasses.”** to place the AQS protocol in context.

Q5. The interpretation of a sharp transition in Fig. 1 is nice; still, this is a nonequilibrium transition. Can the authors use their data to elucidate the nonequilibrium nature of the transition? E.g. dissipation is a central property of nonequilibrium, with the dissipated energy reflected in the area of the stress-strain loop in Fig. 1a. How is this related to the “Landau-like” bimodal energy landscape? How do these two “steady-state” minima arise? (Probably combination of elastic and plastic straining in the oscillatory cycle?)

Indeed, what we study is a nonequilibrium transition in a driven system. The dissipation is indeed captured by the finite areas of the areas enclosed by the stress-strain loops in Fig. 1a. The bimodality of the energy curves is indeed related, since the emergence of two energy minima corresponds to the occurrence of plasticity, which makes the zero stress (and minimum energy) states differ from the zero strain configurations. However, we do not at present have an equilibrium like description of the process wherein the observed behaviour could be explained in terms of transitions between free energy minima. We hope that such a description becomes available in the future.

Q6. In fact, there is a series of experiments by Rogers et al., J. Rheology (e.g. J. Rheol. 55, 435 (2011)) that apply precisely the same analysis of plotting σ_{max} values from the Lissajous figures as done in Fig. 1a and b. How do the current simulations compare to these measurements. Was a similarly sharp drop of σ_{max} as that in Fig.1b observed already in these experiments? The authors should refer, at least briefly, to these measurements.

We have studied the work the referee suggests, and while there are many interesting similarities of approach, we do not find comparable data for the maximum generated stress in the relevant range of strain values. However, we judge the work mentioned to be relevant for our discussion

and have now cited the paper.

Q7. The authors talk about “sluggishness” of the response upon approaching γ_y . When they mention it on page 3, it is unclear what they actually mean. I suggest rephrasing this to clarify the meaning. Later in Fig. 1d it becomes clear that the system close to γ_y needs some time (cycles) to decide where to go; while this makes intuitive sense, what does it actually mean physically? In an equilibrium transition, nucleation plays a central role in delaying the onset of the transition. Is there some similar “nucleation” happening here in the nonequilibrium case? What does the spatial distribution look like; is it localised?

The referee is asking very pertinent and interesting questions, related also to Q5, but at the moment, it is difficult to present any coherent answers. The nucleation picture suggests itself, and is very tempting. We are presently exploring approaches to describing these observations, which we hopefully will be able to do in a publication in the near future. For the moment, we only clarify the meaning of “sluggishness” as follows:

On page 4, we make changes from “..the approach to the steady state becomes increasingly sluggish,..” to “.. **“the number of cycles needed to reach the steady state becomes increasingly large,..”**“

Q8. While I generally believe the author’s interpretation of a sharp transition, some of the graphs do not look like sharp step functions. E.g. if I look at Fig. 1e this could as well be a continuous transition (critical point) entering a “two phase regime”. This is also true for the avalanche sizes in Fig. 2c. This certainly can have to do with the still limited system size, and the increasing sharpness with system size in Fig. 2c seems to point in this direction; still, there is no real proof that the transition itself is sharp. I agree that the qualitative change in behaviour points towards a “coexistence” regime; but couldn’t this be entered through a critical point? In this sense, I would have liked to see the data points for $N > 64000$ that I see in the legend, but unfortunately not in the relevant regime in the graph in Fig. 2c. Maybe they are hidden behind the inset? As this is such a central point of debate in the community right now, it would be valuable to know how sharp the transition really is.

We believe that the totality of the results we present do indicate a sharp transition, in the sense that there is a clear, qualitative change in behaviour across a critical strain amplitude. However, whether this transition is “first order” like or a continuous transition is a question that goes beyond what we are able to answer unambiguously, without stretching the interpretation of our results. We believe we have made the most judicious statements regarding our results in the manuscript. While all the results suggest a sharp transition, which is discontinuous by some measures, it is fair to say this question needs further investigation. The numerical results that we present are at the limit of our capabilities to generate. This is also the reason why we do not have the largest system size data very close to the transition. We performed the $N > 64000$ simulations to confirm the scaling with system size above and below the transition, but the behaviour near the transition will require a much larger numerical investigation.

Q9. Citing only theoretical work on p. 6 is misleading (“... which may be consistent with the suggestion that yielding is a first-order transition [14].”) There is earlier experimental work reporting a sharp “yielding” transition in oscillatory rheology, see in particular Denisov et al., Scientific Reports 5, 14359 (2015). This should be cited here, as it is most relevant and indeed in excellent agreement with the authors’ results.

We thank the referee for bringing this to our attention. In the revised manuscript, we cite this work which is of excellent agreement with our results.

We now cite work of Denisov *et al.* in connection with the possibility of a first order transition in the concluding section.

Referee-3 Comments

A. In this paper the authors study numerical the on-set of the yielding transition using a novel technique: the oscillatory shear deformation. Their main result is the presence of a sharp transition with size dependent avalanches above threshold and localised avalanches below.

B. the technique is novel and allows to target the steady state even below threshold where most of the studies focus on a transient regime.

C-D. the method and the quality of the data appear at the state of the art of the domain

E. I am totally convinced about the existence of the transition and the presence of size dependent avalanches above threshold. I am still surprised about the fact that the mean size of the avalanches does not diverge upon approaching the yielding transition. Is this divergence hidden by the novel protocol that mix avalanches at different stress value? In the further investigation on this point are needed.

It is not the case that the avalanches are mixed at different stress values. Indeed, we have shown evidence that the character of the avalanches resolved by strain (which also means stress below the transition) are the same for different strain windows. This may be the reason for the question of the referee. However, the important point is that the average sizes of the avalanches remain finite regardless of how we analyze them. We believe that the answer to the referee's question lies in the character of the steady states reached under cyclic deformation. As shown in Fig. 1(f), the energies of the cyclically deformed configurations become lower and lower as the transition is approached, and this corresponds also to be absence of large avalanches. We are at present analyzing the manner in which the avalanche sizes decrease with the number of cycles, which we hope to present in a forthcoming publication.

F. The paper is well written and self-contained. I think it deserve publication in Nature Communication

G. The references are appropriate

H. The abstract, the introduction and the conclusion are very clear

We thank the referee for the positive assessment of our work.

Reviewers' Comments:

Reviewer #2 (Remarks to the Author):

The authors have carefully and convincingly addressed all my comments and concerns, and I'm happy to recommend publication of this interesting work.

Reviewer #3 (Remarks to the Author):

I am happy with this revised version and I recommend the publication of this work